# Beyond facility-based births: Is Uganda delivering effective maternal and newborn care? An analysis of the 2022 demographic health survey and 2023 harmonized health facility assessment survey

Brian Turigye[1]*, Edgar Mugema Mulogo[1], Joseph Ngonzi[2], Peter M. Macharia[3,4], Miriam Acheng[5], Aliki Christou[3], Lenka Beňová[3]

1 Department of Community Health, Faculty of Medicine, Mbarara University of Science and Technology, Mbarara, Uganda, 2 Department of Obstetrics and Gynaecology, Faculty of Medicine, Mbarara University of Science and Technology, Mbarara, Uganda, 3 Department of Public Health, Institute of Tropical Medicine, Antwerp, Belgium, 4 Population & Health Impact Surveillance Group, Kenya Medical Research Institute-Wellcome Trust Research Programme, Nairobi, Kenya, 5 Division of Health Information Management, Ministry of Health, Kampala, Uganda

---These authors contributed equally to this work.
* turigyebrian@gmail.com

## Abstract

Maternal and newborn studies in Uganda have primarily focused on measuring contact coverage, such as the proportion of facility-based births. However, this is inadequate and may overestimate the benefits of services provided to women and newborns if the quality of care in the facilities is not considered. Effective coverage of care addresses this limitation and adjusts for the quality of services. This study assessed the effective coverage of maternal and newborn care for facility-based births in Uganda using the 2022 Uganda Demographic and Health Survey (DHS) and the 2023 Harmonized Health Facility Assessment (HHFA). The analysis included 5,618 women who had a live birth in the two years preceding the DHS, and 636 facilities providing childbirth care from the HHFA. Facility readiness was assessed using four domains: human resources, equipment, amenities, and drugs and supplies. Crude coverage was calculated as the percentage of facility births. Two measures of effective coverage were estimated: intervention coverage as a percentage of women who received all ten selected recommended interventions for their most recent birth, and readiness-adjusted coverage as a product of crude coverage and facility readiness using an ecological linking method by region. 85.9% of the women gave birth in a facility, but only 14.0% received all ten interventions. Readiness was highest in government hospitals (81.9%) and lowest in lower government health centers (46.4%). Only 47.8% of women gave birth in a ready facility. Readiness-adjusted coverage varied across regions, with the lowest in Kampala (40.9%) and the highest

**Data availability statement:** The data used for this study were from the Uganda 2022 Demographic and Health Surveys (DHS) and the 2023 Health Facility Assessment (HHFA). The data were obtained from the Uganda Bureau of Statistics (UBOS) and the Uganda Ministry of Health (MOH). These datasets are not publicly available due to ethical and legal restrictions related to participant confidentiality. However, de-identified data may be made available to interested researchers upon reasonable request from the relevant authorities. Access to the DHS dataset can be requested through the Uganda Bureau of Statistics (UBOS) by contacting info@ubos.org or visiting https://www.ubos.org. Access to the HHFA dataset can be requested from the Uganda Ministry of Health (MOH) by contacting ps@health.go.ug.

**Funding:** BT received a PhD scholarship from VLIR-UOS through the MUST-UCOBS project, which also supported his attendance at the DHS course at the Institute of Tropical Medicine (ITM). PMM is supported by a Senior Postdoctoral Fellowship from the Fonds voor Wetenschappelijk Onderzoek (FWO), Belgium (Grant number: 1201925N). The funders had no role in study design, data collection and analysis, decision to publish, or preparation of the manuscript.

in the North-Eastern (61.4%). These Findings indicate a large gap between crude and effective coverage, disproportionately affecting regions and lower-level health centers, highlighting a need to enhance the capacity of lower-level health centers to deliver quality maternal and newborn care.

## Introduction

For decades, maternal and newborn health have been a global public health priority [1]. This commitment is emphasized by the Sustainable Development Goal targets (SDG) 3.1 and 3.2, which aim to reduce maternal, newborn, and under-five mortality rates by 2030 [2]. Despite these efforts, maternal and newborn mortality rates remain unacceptably high, particularly in sub-Saharan Africa and Southern Asia. Every day, 700 women worldwide die due to pregnancy-related complications and childbirth, and 92% of these occur in low-and middle-income countries [3]. Worldwide, the first month of life is the most vulnerable for children under five, with about 2.3 million newborn deaths and an estimated 1.9 million stillbirths annually [4]. According to the 2022 estimates, Uganda has a maternal mortality ratio (MMR) of 189 deaths per 100,000 live births—significantly above the SDG target of 70 and the supplemental threshold of 140 that no country should exceed by 2030. Similarly, Uganda's neonatal mortality rate of 22 deaths per 1,000 live births is nearly double the global SDG target of 12 by 2030 [5].

Many of these maternal and newborn deaths and stillbirths are preventable if access to high-quality care during pregnancy and childbirth is ensured [6,7]. In light of this, global and local efforts have been geared towards improving access to maternal and newborn interventions and services such as skilled birth attendance and facility-based antenatal and childbirth care [8]. Uganda has made significant strides in improving health service utilisation for pregnant and birthing women; for example, utilisation of health facilities for childbirth increased from 44% in 2001 to 86.4% in 2022 [5,9,10].

However, utilisation of these services alone is not enough to reduce the maternal and newborn deaths [11]. Studies worldwide have often concentrated on measuring coverage and utilisation of maternal and newborn care services [12]. There is now increasing evidence that this overestimates the benefits of the service if the quality of the service is not taken into consideration [13]. Consequently, there have been increasing calls to shift from just measuring crude coverage to effective coverage, which accounts for the quality of the service provided [14,15]. Effective coverage moves beyond measuring contact with a service to include the element of quality or standards of care needed to achieve a positive health outcome [15,16].

Effective coverage has been described along a cascade, including the target population (population in need of health service), contact coverage (proportion that contacts health service), input-adjusted/readiness-adjusted coverage (proportion that contacts a ready health service), intervention coverage (proportion that receives the recommended health service), quality adjusted coverage (proportion that receives

service according to standards), and outcome adjusted coverage(proportion with positive outcomes) [15]. While there has not been a single standard definition of effective coverage for childbirth care, input-adjusted coverage, intervention coverage, and quality-adjusted coverage have been recommended as proxy measures for effective coverage in maternal and newborn care. Therefore, the three parameters have been recommended as the preferred measures for maternal and newborn care services and were the focus of this study [15]. This is because the outcome measures - maternal and newborn mortality - are influenced by a multitude of factors, some outside the health system, and it may therefore be difficult to attribute them to the healthcare system gaps [15,17,18]. To measure effective coverage, most studies have used household survey data and health facility assessment data [17].

Globally, previous studies have documented low effective coverage despite the high percentage of facility births. For instance, a scoping review in low- and middle-income countries found the effective coverage for facility births to be as low as 0%, while the crude coverage of facility births was reported to be as high as 90% [19]. However, no prior national-level studies have examined the effective coverage of maternal and newborn care services for facility-based births in Uganda. To better take into account the importance of the environment in which childbirth care is provided and the provision of recommended interventions around childbirth, this study focuses on crude coverage, readiness-adjusted coverage, and intervention coverage for facility-based births in Uganda using recent population- and health facility-based surveys. The objective of this study was to assess the effective coverage of maternal and newborn care for facility-based births in Uganda.

## Methods

### Study setting

Uganda is located in the Eastern part of Africa and consists of nine geographical subregions (Fig 1). The country's healthcare is delivered by both the public (government) and the private sectors. The private sector is composed of both private for-profit and private not-for-profit (PNFP) entities and constitutes 55% of health facilities nationally [20]. Public healthcare is delivered through a devolved, decentralized network of facilities consisting of the national centralized health system and the district health system [21]. The central health system includes the National Referral Hospital (NRH) and Regional Referral Hospitals (RRH), with the majority concentrated in urban areas, either the capital Kampala or regional capitals.

The district health system, which is further divided into health subdistricts, consists of district general hospitals, Health centres (HC) IV, III, and II, while village health teams support care within the communities. HCIIs offer antenatal, family planning, and immunization services, but care during childbirth is provided from the HCIII level and above [20]. HCIVs and above should be functioning as Comprehensive Obstetric and Newborn Care (CEmONC) facilities with all signal functions [22]. The central health system facilities are referral points for the lower district health system facilities [20,23].

### Data sources

We used secondary data from two surveys: the most recent nationally representative Demographic Health Survey (DHS) of 2022 and the Harmonised Health Facility Assessment Survey(HHFA) of 2023.

The 2022 Uganda DHS was a cross-sectional household survey conducted by the Uganda Bureau of Statistics (UBOS). The survey used a two-stage stratified sampling design. First, a list of enumeration areas (EAs) was generated from the 2014 National Population and Housing Census to generate a sampling frame. In the first stage, a total of 697 EAs were sampled with probability proportional to size across all 14 regions of the country. Households were then listed in each of the 697 EAs, and this sampling frame was used to randomly select 30 households from each. All households were included for EAs that had fewer than 30. We used the household and women's questionnaires for this study. A total of 20,481 households were sampled in the 2022 UDHS survey, within which 18,251 women of reproductive age were approached [5].

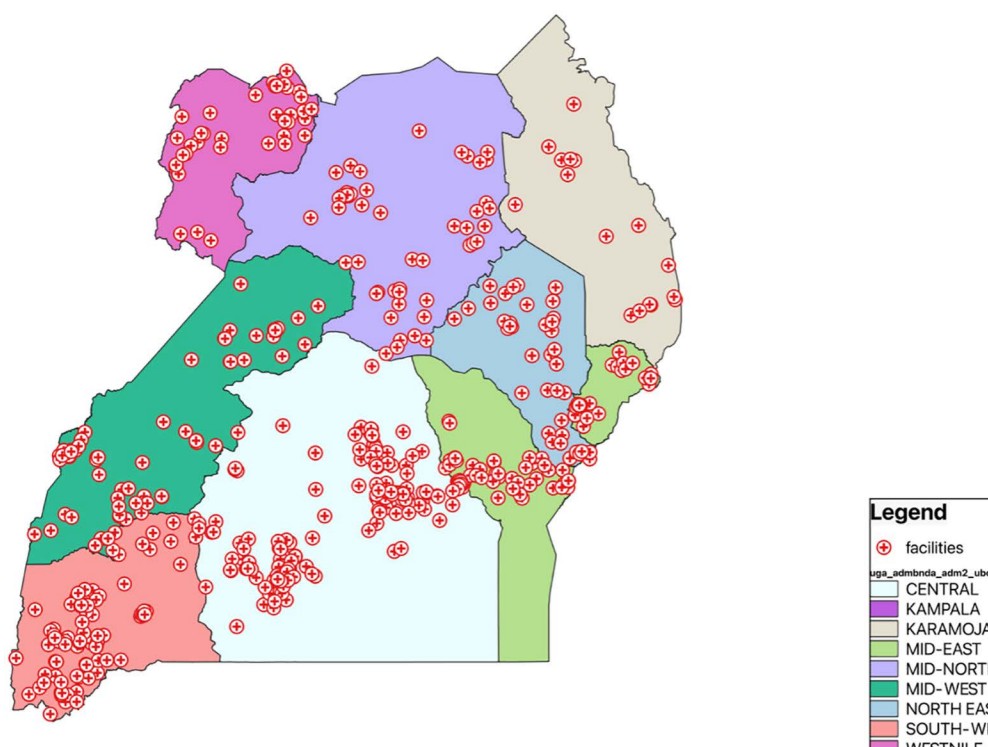

**Fig 1. A map of Uganda showing the study sub-regions and the geographical distribution of facilities sampled in the HHFA.** The base map was created in QGIS using administrative boundary data from the Humanitarian Data Exchange (https://data.humdata.org/dataset/cod-ab-uga), licensed under CC BY 4.0.

The HHFA was a nationwide health facility-based survey that assessed the ability and readiness of health facilities to provide general and maternal/newborn health services. The survey measured service readiness indicators, including availability of trained staff, guidelines, equipment, laboratory services, and medicines. The Uganda 2023 HHFA was conducted in April 2022 across the country by the Ministry of Health (MOH). The assessment utilised a stratified equal probability systematic sampling technique using a national health facility master list generated by the MOH. First, the country was divided into 15 geographical subregions and 2–3 districts randomly selected from each subregion. At the second stage, in each sub-region, the lower-level health facilities were stratified by level (HC IV, III, and II) and managing authority (public, private-for-profit, and PNFP), and a systematic random sampling technique was used to select facilities until the required sample size was reached. All national, regional, referral, and general hospitals were purposively included. A standard WHO HHFA questionnaire was used and administered using interviews with unit in charges or a delegated staff and physical inspection. A total of 636 facilities were surveyed [24]. Since the two data sets had distinct geographical categories, we harmonised them into nine subregions for the purpose of analysis.

as shown in Fig 1.

## Study population

The study population was women aged 15–49 years with a live birth in the two years preceding the 2022 DHS data (target population). We restricted our sample to the most recent live birth in the two years preceding the survey because our intervention coverage indicators were only collected for the most recent birth, and to maximise recall [25,26]. In the case of multiples, the lastborn infant was taken into account (one newborn per woman).

## Definitions

We measured effective coverage using four parameters of a cascade proposed by the Effective Coverage Think Tank Group [15]. These are illustrated in **Fig 2** as: target population, crude/contact coverage, readiness/input-adjusted coverage, and intervention coverage.

The target population was used as the denominator for all the effective coverage measurements.

We defined crude/contact coverage as the percentage of the target population who gave birth in a health facility. Health facilities were categorised into government hospital, government health centre, and private facility in alignment with the HHFA categories described in detail in S1 Table.

We defined readiness/input-adjusted coverage as the percentage of the target population that gave birth at a ready health facility. We defined readiness using health facility data from the 2023 HHFA survey. This survey collected comprehensive data on all health services, some of which may not be relevant for maternal and newborn care. We used the WHO QMNC framework guidelines to choose 36 key indicators. These were divided into four domains: 1) Equipment, 2) Basic amenities, 3) Staff availability and training, and 4) Drugs and supplies [27]. Each domain had a set of indicators. Facility readiness was scored on a scale from 0 to 1. A score of 1 meant the indicator was available, while 0 meant it was not. This gave us an availability score (see S2 Table for all the indicators used to calculate the readiness scores of each domain).

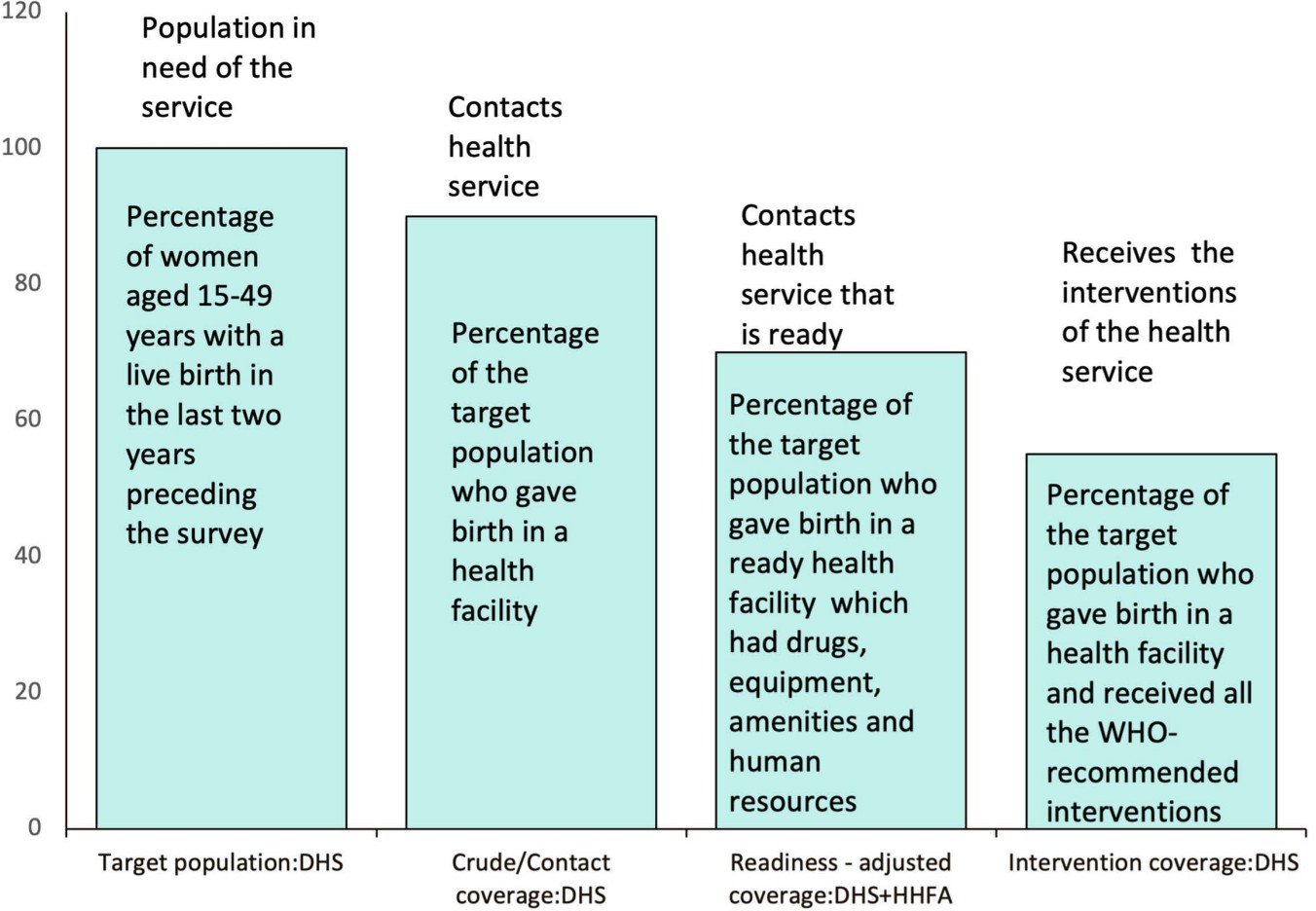

**Fig 2. Effective coverage cascade of maternal and newborn care. Adapted from [15].**

Using the DHS methodological report on quality of care, we adopted the simple additive method to calculate readiness. This approach was chosen to ensure that each indicator contributes equally to the overall readiness irrespective of the domain [28–30].

$$Readiness\ index\ R(x) = \frac{(Total\ number\ of\ "yes"\ responses\ across\ all\ domains)X\ 100}{Total\ number\ of\ indicators\ across\ all\ the\ domain}$$

We used an ecological approach to link the women of reproductive age who had a live birth in the past two years from the DHS data to Health facilities based on the regional location of these facilities in the HHFA data. While several linking approaches have been suggested, this method has been widely validated for linking DHS data with Health facility assessments (including the HHFA) [31,32]. We therefore linked these data sets by region. The two surveys collected data for regions and facility types differently; we therefore first harmonized the response options for the two variables, and the details are described in (S1 Table)

Intervention Coverage was defined as the percentage of the target population who gave birth in a health facility and received all the selected WHO-recommended interventions.

$$Intervention\ coverage = \frac{Number\ of\ women\ with\ a\ live\ birth\ in\ the\ last\ two\ years\ who\ received\ all\ the\ selected\ interventions}{Total\ women\ a\ with\ a\ live\ birth\ in\ the\ two\ years\ (target\ population)}$$

We did not restrict our target population to only women who delivered from "ready" facilities to make this a nested cascade that conditions receiving interventions to only those gave birth at ready facilities from the previous step. We therefore calculated intervention coverage independently from readiness-adjusted coverage as two separate approaches for effective coverage measurements. This was because even facilities with poor readiness scores may still be capable of delivering these interventions, and we wanted to capture the coverage of the recommended interventions among the women in need. Our effective coverage measure set out to assess not only the proportion of individuals in need of facility-based services who received them (crude/contact coverage), but also the selected recommended interventions they received (intervention coverage) based on similar studies that have proposed or used this approach [17,33–35]. We, however, restricted receiving interventions to only women who gave birth in health facilities. Therefore, all women who did not give birth at a health facility were considered as not having received any intervention.

We selected 10 interventions based on immediate post-partum care recommendations from earlier studies [29,30,36] and the interventions recommended within the WHO Quality of Maternal and Newborn Care (QMNC) framework, which were available in the 2022 Uganda DHS dataset [27]. These interventions included: bleeding checked, woman's first post-natal care (PNC) check within 48 hours, blood pressure was taken during the PNC check, newborn's First PNC check within 48 hours, newborn temperature measured, skin to skin care, cord examined, observed breastfeeding, counselling on danger signs of newborn, counselling on breastfeeding

We included the following variables to describe the background characteristics of women and the care they received, which were selected based on availability and relevance to the study objectives: region of residence, maternal age, marital status, parity, residency, wealth index, maternal education level, religion, pregnancy intention, mode of birth, and antenatal care (ANC) attendance. The details of how variables were categorised and treatment of missing values are available in (S3 Table).

## Data analysis

Data was analysed using STATA version 18.5. For DHS data, weighted data analysis was done to account for the multistage stratified sampling design by applying standard sampling weights, strata, and cluster variables already generated in the dataset by the DHS program and UBOS. We used the svyset command to apply this weighting [37]. No weights were applied for analysing HHFA data or ecological linking. We conducted descriptive statistical analysis and presented results as percentages, using tables and graphs to describe the crude coverage, input/readiness-adjusted coverage, and intervention coverage.

We calculated 95% confidence intervals (CIs) for the outcome variables. To calculate readiness-adjusted coverage, adopted an approach utilized in previous studies [35,38], by multiplying the crude/contact coverage with the readiness scores, applying the delta approach presented as a percentage with uncertainty intervals [39]. Data was analysed and presented by region.

For both DHS and HHFA data, women or health facilities with missing values or responses other than "yes" to any of the questions measuring the outcome variables on effective coverage or availability of a readiness parameter were considered as no. Details on how missing values for each variable were treated are in the S1, S2, and S3 Tables.

## Ethics Statement

This study utilised publicly accessible data and therefore did not require Ethical approval. Authorisation and access to UDHS data were granted by UBOS, while HHFA data access was granted by the Uganda MOH. The UDHS data was accessed from March 27th, 2025, while the HHFA was accessed from May 24th, 2025.The authors were not involved in data collection and did not have access to any identifying information for study participants.

## Results

### Socio-demographic characteristics and distribution of births by location

Table 1 summarises the study sample of women and distribution of births by location according to women's background characteristics. Of the 5,618 women surveyed, the majority resided in rural areas (70.4%), completed primary education (58.1%), delivered vaginally (86.3%) and attended four or more ANC visits (73.1%).

Overall, 85.9% (95% CI: 84.9-86.9) of women gave birth in a health facility. We disaggregated births by location and found that the largest percentage took place in government health centres (47.0%), followed by government hospitals 26.4%, while the least were in private facilities (12.6%), and 14.1% were home births.

Facility births were highest among urban (93.0%), primipara (93.4%), the highest wealth quantile (96.7%), higher education (95.9%), and those who attended four or more ANC visits (74.8%). Only 0.8% of women with no ANC gave birth at a health facility. Kampala region had the highest percentage of facility births (95.6%) and Karamoja the lowest (80.2%). The majority of births in Kampala (49.4%) and the central region (44.0%) took place in government hospitals, which differed from all other regions, where a majority of births were in government health centres. These two regions also had the highest percentage of private facility births, with Kampala at 22.3% and the Central region at 21.6%.

### Health facility readiness to provide maternal and newborn care

The readiness of facilities to provide maternal and newborn care from the HHFA is described in Table 2 and presented by region and facility type. A total of 636 facilities were surveyed; the largest number (23.3%;n=148) were located in the central region, while Karamoja had the fewest (2.5%; n=16). Most of the surveyed facilities were government health centers (45.4%; n=289), followed by private facilities (43.2%; n=275), and government hospitals (11.3%; n=72). Overall, the percentage of facilities deemed ready to provide childbirth care was 55.7% (95% CI: 53.1, 58.3), with readiness ranging from 42.8% (95% CI: 35.7, 50.1) in Kampala to 71.8% (95% CI: 62.2, 81.2) in West Nile. Government hospitals had the highest level of readiness at 81.9%, followed by private facilities at 58.7%. In contrast, government health centers demonstrated the lowest readiness level at 46.4%.

### Readiness-adjusted/effective coverage of maternal and newborn care

To estimate effective coverage, we adjusted the regional and overall crude coverage by facility readiness. The overall readiness-adjusted coverage/effective coverage was 47.8% (95% CI 45.5,50.1). This was highest in Westnile 61.4% (95% CI; 51.3,71.5) and lowest in Kampala 40.9% (95% CI; 33.9,47.9) (Table 3).

- For each region's effective coverage estimation, we used the overall readiness score and crude coverage for that region

**Table 1. Background characteristics and distribution of births among women with live births in the 2 years preceding the survey, by place of birth in Uganda (2022 UDHS).**

| Variables | Percentage of women with a live birth in the two years preceding the survey (Column %) n (%) | | Distribution of births by place (*Row %*) | | | | Percentage of women with a live birth in the last two years who gave birth at a health facility (B+C+D) (%, 95% CI) Crude/contact coverage) |
|---|---|---|---|---|---|---|---|
| | | | Home (A) | Gov. hospital (B) | Gov. health centre (C) | Private facility (D) | |
| **Region** | | | | | | | |
| Kampala | 249 | (4.4) | 4.4 | 49.4 | 23.9 | 22.3 | 95.6 [92.1,97.5] |
| Central | 1,333 | (23.7) | 10.1 | 44.0 | 24.4 | 21.6 | 89.9 [86.8,92.3] |
| Karamoja | 390 | (6.9) | 19.8 | 15.0 | 64.2 | 1.0 | 80.2[72.2,86.4] |
| Mid-North | 597 | (10.6) | 15.2 | 11.4 | 62.6 | 10.8 | 84.8[79.2,89.0] |
| Mid-East | 748 | (13.3) | 15.4 | 17.1 | 51.4 | 16.1 | 84.6 [80.9,87.7] |
| North-East | 749 | (13.3) | 18.2 | 32.1 | 48.0 | 1.7 | 81.8[77.8,85.3] |
| Mid-West | 771 | (13.7) | 16.9 | 17.4 | 55.1 | 10.7 | 83.1[79.5,86.3] |
| South-West | 566 | (10.1) | 11.4 | 19.1 | 57.9 | 11.6 | 88.6[85.2,91.2] |
| Westnile | 215 | (3.8) | 14.5 | 17.4 | 62.3 | 5.8 | 85.5[75.1,92.0] |
| **Age (in years)*** | | | | | | | |
| ≤19 | 1,041 | (18.5) | 11.5 | 27.9 | 50.4 | 10.2 | 88.5[86.2,90.5] |
| 20-24 | 1,685 | (30.0) | 14.5 | 27.3 | 45.3 | 12.9 | 85.5[83.4,87.5] |
| 25-34 | 2,199 | (39.1) | 13.1 | 25.9 | 47.7 | 13.2 | 86.9[84.9,88.6] |
| 35-49 | 693 | (12.3) | 20.1 | 23.3 | 43.5 | 13.1 | 79.9[76.1,83.2] |
| **Marital status** | | | | | | | |
| Never married | 336 | (6.0) | 10.4 | 30.5 | 44.7 | 14.5 | 89.6[85.4,92.7] |
| Married | 2,414 | (43.0) | 14.2 | 26.4 | 47.4 | 12.1 | 85.8[83.7,87.7] |
| Living with partner | 2,255 | (40.1) | 14.2 | 25.5 | 47.1 | 13.2 | 85.8[83.9,87.5] |
| Widowed/Separated | 613 | (10.9) | 15.4 | 27.5 | 45.9 | 11.1 | 84.6[81.3,87.4] |
| **Highest education achievement** | | | | | | | |
| No education | 475 | (8.5) | 24.7 | 13.1 | 56.3 | 5.9 | 75.3 [69.7,80.2] |
| Primary | 3,264 | (58.1) | 16.6 | 23.9 | 49.9 | 9.6 | 83.4[81.6,85.0] |
| Secondary | 1,611 | (28.7) | 7.5 | 34.0 | 39.9 | 18.6 | 92.5[90.8,94.0] |
| Higher | 268 | (4.8) | 4.1 | 34.1 | 37.3 | 24.5 | 95.9 [92.0,98.0] |
| **Religion** | | | | | | | |
| Christian | 4,819 | (85.8) | 15.0 | 25.0 | 48.1 | 11.9 | 85.0[83.5,86.4] |
| Muslim | 746 | (13.3) | 7.7 | 36.5 | 38.5 | 17.3 | 92.3[89.6,94.3] |
| Others | 53 | (1.0) | 19.7 | 13.2 | 57.9 | 9.2 | 80.3[67.5,88.9] |
| **Type of Residence** | | | | | | | |
| Urban | 1,661 | (29.6) | 6.8 | 42.8 | 33.7 | 16.7 | 93.2[91.3,94.8] |
| Rural | 3,957 | (70.4) | 17.2 | 19.5 | 52.5 | 10.8 | 82.8[81.1,84.5] |
| **Wealth index** | | | | | | | |
| Q1. Lowest | 1,276 | (22.7) | 22.1 | 15.5 | 56.0 | 6.4 | 77.9[74.7,80.8] |
| Q2. Second | 1,117 | (19.9) | 19.5 | 19.0 | 54.7 | 6.7 | 80.5 [77.5,83.2] |
| Q3. Middle | 1,016 | (18.1) | 13.8 | 23.5 | 49.5 | 13.1 | 86.2[83.6,88.4] |
| Q4. Fourth | 1,083 | (19.3) | 10.5 | 29.5 | 45.1 | 14.8 | 89.5[87.1,91.5] |
| Q5. Highest | 1,126 | (20.0) | 3.3 | 45.6 | 28.4 | 22.7 | 96.7[95.3,97.7] |

*(Continued)*

**Table 1.** (Continued)

| Variables | Percentage of women with a live birth in the two years preceding the survey (Column %) n (%) | | Distribution of births by place (*Row %*) | | | | Percentage of women with a live birth in the last two years who gave birth at a health facility (B + C + D) (%, 95% CI) Crude/contact coverage) |
|---|---|---|---|---|---|---|---|
| | | | Home (A) | Gov. hospital (B) | Gov. health centre (C) | Private facility (D) | |
| **Parity** | | | | | | | |
| Primipara | 1,320 | (23.5) | 6.6 | 32.2 | 47.8 | 13.4 | 93.4[91.8,94.7] |
| 2-4 | 2,730 | (48.6) | 14.5 | 26.5 | 45.9 | 13.1 | 85.5[83.6,87.2] |
| ≥5 grand multipara | 1,568 | (27.9) | 19.6 | 21.4 | 48.1 | 10.9 | 80.4[77.7,82.7] |
| **Mode of Childbirth** | | | | | | | |
| Vaginal delivery | 4,851 | (86.3) | 16.3 | 22.6 | 49.5 | 11.6 | 83.7 [82.1,85.2] |
| C-section | 767 | (13.7) | 0 | 50.5 | 30.8 | 18.8 | 100.0 |
| **Sex of the newborn** | | | | | | | |
| Male | 2,812 | (50.1) | 13.7 | 26.7 | 46.5 | 13.1 | 86.3[84.6,87.9] |
| Female | 2,806 | (49.9) | 14.5 | 26.1 | 47.5 | 12.0 | 85.5[83.7,87.1] |
| **Pregnancy intention#** | | | | | | | |
| Wanted | 3,688 | (65.7) | 12.9 | 27.8 | 46.1 | 13.2 | 87.1[85.5,88.5] |
| Not wanted | 1,930 | (34.3) | 16.3 | 23.8 | 48.6 | 11.4 | 83.7[81.4,85.7] |
| **Number of ANC visits** | | | | | | | |
| None | 65 | (1.2) | 37.6 | 25.6 | 24.0 | 12.8 | 0.8 [0.6,1.2] |
| 1-3 | 1,447 | (25.8) | 18.8 | 25.8 | 42.4 | 13.1 | 24.4[22.8,26.0] |
| 4+ | 4,106 | (73.1) | 12.1 | 26.6 | 48.9 | 12.4 | 74.8 [73.2,76.4] |
| **TOTAL** | **5,618** | **(100)** | **14.1** | **26.4** | **47.0** | **12.6** | **85.9[84.9, 86.9]** |

\* Refers to maternal age at the time of delivery.

\# Refers to the wantedness of the pregnancy at the time of conception.

Figures represent weighted data, and counts were rounded off to the nearest whole number.

**Abbreviations:** ANC: Antenatal Care, Gov.: Government.

### Intervention coverage of maternal and newborn care

**Table 4** describes the receipt of the selected ten care interventions for childbirth, which form the basis of the estimate of intervention coverage. The most covered intervention was immediate skin-to-skin contact, which was done in 84.9% of all births, while the least completed intervention was measuring blood pressure during PNC, which was performed in only 32.4% of mothers. The intervention coverage among women with a live birth in the two years preceding DHS (all interventions received) was 14.0% (95% CI: 12.6,15.5) overall. This ranged from a low of 5.0% (95% CI: 3.4,7.2) in the North-East to a high of 31.7% (95% CI: 25.1,39.1) in Karamoja

### Cascade for effective coverage of childbirth care

Of the 5618 (100%) women surveyed/target population, 85.9% gave birth in a health facility (crude coverage). We estimated that 47.8% gave birth in a facility that was ready to provide such care (readiness-adjusted coverage). However, only 14% of women with a live birth in the 2 years preceding DHS reported receiving all the interventions (intervention coverage) (**Fig 3**).

**Table 2. Health facility Readiness to provide quality maternal and newborn care by region and facility type from the HHFA survey, n = 636 facilities.**

| Variables | Number of health facilities per region | | Readiness/percentage of facilities with all the indicators available or ready on the day of the survey | Readiness score by type of facility | | |
|---|---|---|---|---|---|---|
| Region | | | Overall* | Government Hospital n = 72 (11.3%) | Government Health centre n = 289 (45.4%) | Private facility n = 275 (43.2%) |
| | n | % | | | | |
| Kampala | 99 | 15.6 | 42.8[35.7,50.1] | 52.3 | 19.0 | 43.8 |
| Central | 148 | 23.3 | 56.3[50.9,61.8] | 82.8 | 43.7 | 64.0 |
| Karamoja | 16 | 2.5 | 67.5[50.2,84.9] | 94.5 | 52.5 | 90.7 |
| Mid-North | 51 | 8.0 | 59.4[50.9,68.1] | 91.6 | 46.4 | 75.7 |
| Mid-East | 74 | 11.6 | 57.7[50.1,65.2] | 87.3 | 47.5 | 67.4 |
| North-East | 56 | 8.8 | 50.6[42.8,58.3] | 76.6 | 44.6 | 47.0 |
| Mid-West | 69 | 10.9 | 60.8[53.6,67.9] | 81.7 | 48.3 | 70.2 |
| South-West | 83 | 13.1 | 55.2[47.9,62.5] | 81.5 | 44.9 | 63.6 |
| Westnile | 40 | 6.3 | 71.8[62.2,81.2] | 96.7 | 60.1 | 79.0 |
| TOTAL | 636 | 100 | 55.7[53.1, 58.3] | 81.9 | 46.4 | 58.7 |

**Table 3. Readiness -adjusted/effective coverage of maternal and newborn care by region.**

| Variable | Number of women with a live birth in the last two years | Percentage of women with a live birth in the last two years who gave birth in a health facility Crude coverage %[95%CI] | Readiness score %[95%CI] | Crude coverage* x Readiness Effective coverage/readiness adjusted coverage % [Uncertainty interval] |
|---|---|---|---|---|
| Region | | | | |
| Kampala | 249 | 95.6 [92.1,97.5] | 42.8[35.7,50.1] | 40.9[33.9,47.9] |
| Central | 1,333 | 89.9 [86.8,92.3] | 56.3[50.9,61.8] | 50.6[45.5,55.8] |
| Karamoja | 390 | 80.2[72.2,86.4] | 67.5[50.2,84.9] | 54.1[39.4,68.9] |
| Mid-North | 597 | 84.8[79.2,89.0] | 59.4[50.9,68.1] | 50.4[42.5,58.2] |
| Mid-East | 748 | 84.6 [80.9,87.7] | 57.7[50.1,65.2] | 48.8[42.1,55.5] |
| North-East | 749 | 81.8[77.8,85.3] | 50.6[42.8,58.3] | 41.4[34.8,48.0] |
| Mid-West | 771 | 83.1[79.5,86.3] | 60.8[53.6,67.9] | 50.5[44.2,56.8] |
| South-West | 566 | 88.6[85.2,91.2] | 55.2[47.9,62.5] | 48.9[42.2,55.6] |
| Westnile | 215 | 85.5[75.1,92.0] | 71.8[62.2,81.2] | 61.4[51.3,71.5] |
| Total | 5,618 | 85.9[84.9, 86.9] | 55.7[53.1, 58.3] | 47.8[45.5,50.1] |

*Effective coverage was calculated by cross-multiplying point estimates and confidence intervals of the crude coverage and the readiness facility score, adjusting the crude coverage for readiness

*Readiness-adjusted coverage and intervention coverage were calculated as independent measures for effective coverage.*

## Discussion

Measuring only contact coverage without accounting for the quality of care is insufficient to understand the gaps in maternal and newborn care services [13,33,40]. This study, therefore, combined data from Uganda's most recent nationally representative 2022 DHS and 2023 HHFA surveys to estimate the effective coverage of maternal and newborn care for childbirths. We found that while a high percentage of women who gave birth at a health facility, less than half of the

Table 4. Intervention Coverage; 2022 DHS data.

| Variable | Intervention specific coverage; Percentage of women with a live birth in the last two years received an intervention N=5618 | | | | | | | | | | Percentage of women with a live birth in the last two years who received all the selected interventions % [95% CI] |
|---|---|---|---|---|---|---|---|---|---|---|---|
| | Bleeding checked | Woman's 1st PNC check within 48 hours | Blood pressure was taken during PNC check | Newborn First PNC check within 48 hours | Newborn temperature measured | Skin to skin care | Cord examined | Observed breast feeding | Counselled on danger signs of newborn | Counselling on Breastfeeding | ALL INTERVENTIONS (Intervention coverage) |
| | 2,105 (37.5%) | 3,622 (64.5%) | 1,818 (32.4%) | 3,634 (64.7%) | 2,270 (40.4%) | 4,768 (84.9%) | 2,672 (47.6%) | 2,560 (45.6%) | 2,139 (38.1%) | 2,546 (45.3%) | 784 |
| | | | | | | | | | | | 14.0 [12.6,15.5] |
| **Region** | | | | | | | | | | | |
| **Regional coverage of interventions** | | | | | | | | | | | |
| Kampala | 47.6 | 83.4 | 47.7 | 83.7 | 60.8 | 85.0 | 63.6 | 61.1 | 57.0 | 60.2 | 24.8 [19.3,31.2] |
| Central | 44.6 | 74.6 | 44.3 | 73.0 | 51.9 | 84.5 | 54.4 | 52.9 | 46.3 | 50.4 | 21.3 [17.3,25.9] |
| Karamoja | 63.1 | 69.3 | 71.6 | 70.3 | 73.9 | 86.7 | 87.2 | 60.8 | 57.0 | 76.8 | 31.7 [25.1,39.1] |
| Mid-North | 38.9 | 55.4 | 24.4 | 59.0 | 34.9 | 82.9 | 42.1 | 44.0 | 40.5 | 44.7 | 10.3 [7.4,14.1] |
| Mid-East | 35.2 | 63.2 | 25.7 | 62.9 | 36.8 | 87.4 | 42.8 | 42.1 | 36.5 | 40.2 | 10.4 [7.6,13.9] |
| North-East | 17.7 | 49.4 | 13.8 | 51.5 | 24.9 | 85.3 | 34.6 | 33.5 | 22.8 | 34.4 | 5.0 [3.4,7.2] |
| Mid-West | 26.2 | 63.0 | 21.4 | 62.2 | 28.2 | 87.3 | 34.9 | 37.2 | 27.7 | 35.7 | 8.5 [5.7,12.5] |
| South-West | 39.2 | 61.9 | 29.3 | 61.2 | 31.5 | 80.4 | 43.2 | 43.8 | 29.9 | 38.4 | 8.8 [6.3,12.1] |
| Westnile | 43.9 | 64.7 | 26.8 | 67.5 | 34.6 | 82.0 | 48.5 | 47.5 | 41.7 | 49.3 | 11.4 [7.4,17.1] |

Figures represent weighted data, and counts were rounded off to the nearest whole number.

PNC: Postnatal Care.

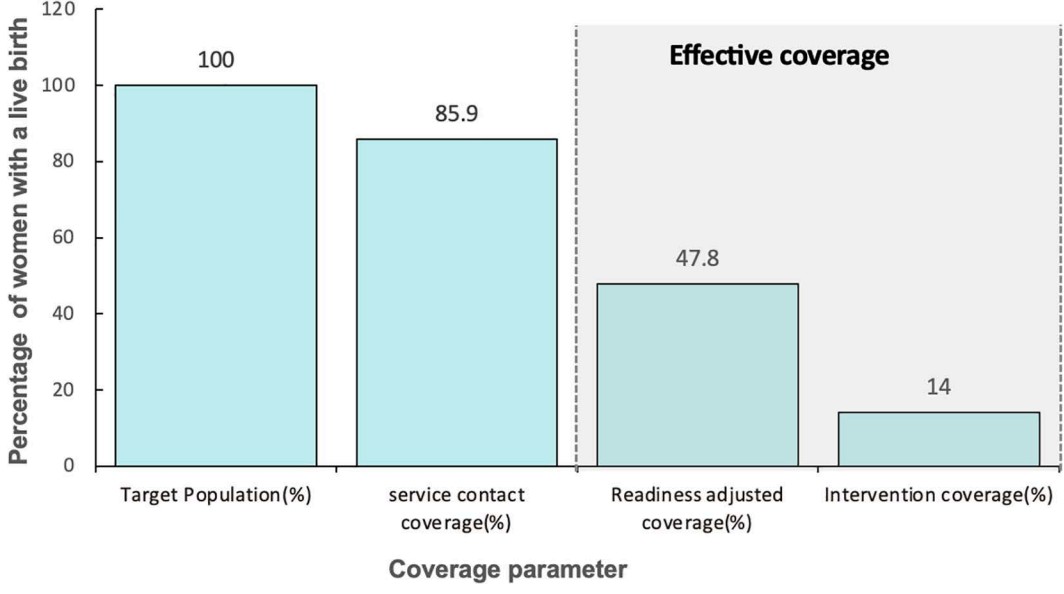

**Fig 3. Effective coverage cascade of childbirth care, Uganda.**

women in need of childbirth care gave birth from health facilities that were ready to provide care, and only 14% gave birth from health facilities and received the selected standard recommended interventions.

Our study identified a substantial gap between contact/crude coverage, readiness-adjusted coverage, and intervention coverage, with majority of the women not receiving the recommended interventions despite giving birth at a health facility. This indicates that while Uganda has made great efforts to increase the percentage of facility births above the current global average of 81% and far above the sub-Saharan Africa average of 64%, it is evident that women do not necessarily receive the required quality care [41,42]. In addition to efforts to promote facility births, there is a need to amplify efforts to improve the quality of care offered in these facilities, which is unacceptably low. Evidence now suggests that there is little or no correlation between facility births and improving maternal or newborn outcomes if the content of care and quality are not addressed [43]. Unfortunately, like many global maternal and newborn programs, there has been a strong emphasis on contact coverage with little emphasis on readiness and the content provided. Encouraging mothers to seek care from health facilities that are unprepared will not improve maternal and newborn outcomes without addressing the gaps in quality of care and can have negative consequences, including loss of community trust and poor experiences of care that deter future care-seeking [44].

It is perhaps not surprising that intervention coverage was low, given that less than half of health facilities included in our analysis met the required human resource standards. This could have partly arisen due to over-stretched healthcare providers or an inadequate number of competent and motivated skilled health workers. Prior studies in Uganda have documented work overload as a major barrier to offering standard care, as health workers are strained to provide the required care and interventions to mothers when insufficient staff are available. Exacerbating the staff shortages amidst overwhelming duties is always a lack of adequate skills and continuous training to offer this care [45–47]. We observed that regions with higher facility readiness did not necessarily have corresponding higher levels of intervention coverage. For instance, although Kampala had the lowest facility readiness, it did not have the lowest intervention coverage. This finding aligns with previous studies from Uganda and similar settings, which have found weak or no consistent association between facility readiness and intervention coverage [48–50]. This may be because intervention coverage is influenced not only by facility readiness, but also by factors related to health workers, individual behaviors, and socio-demographic characteristics [48–50]. These findings suggest that delivering in a well-equipped facility does not always guarantee that mothers will receive all the recommended interventions.

A gap of over 38% between contact coverage and readiness-adjusted coverage is a manifestation of how unprepared health facilities are to deliver quality care, highlighting a missed opportunity for improving effective coverage and maternal or newborn care in Uganda. Readiness was lowest and below average in government health centres, yet the majority, almost half of all the births, took place at these centres, indicating a missed opportunity for improving care in the places where most births take place. This is comparable to related studies in East Africa that have documented disproportionately low readiness in lower health public facilities [51,52]. Also, in agreement with these studies, our study showed that most of the women who gave birth from these government lower facilities were from the poorest quantile and rural areas [44,45]. This finding illuminates existing gaps in achieving universal health coverage for maternal and newborn care in these underserved populations, which is greatly hinged on access to quality care [46]. While Uganda has made significant gains in improving maternal and newborn care, there is still a need to provide more health care resources, especially at lower-level government health facilities [31]. The persistent lack of basic amenities, functional equipment, drug supplies, competent and motivated human resources, as indicated in this study's findings, remains a bottleneck to providing equitable quality care. Our findings contribute to the broader debate on whether delivery care should be decentralized to lower-level facilities or centralized in better-equipped referral hospitals [53–55]. While some studies have questioned the feasibility of equipping all lower-level facilities to provide childbirth care and instead advocate for centralizing services [55], evidence from countries such as Tanzania and Uganda suggests that upgrading selected lower-level facilities to provide CEmONC services is both feasible and impactful [56,57]. In light of these findings from similar settings, our results underscore the urgent need to strengthen the readiness of lower-level government facilities, including the potential upgrading of those providing childbirth services to meet CEmONC standards.

Although the facility readiness was generally poor in all regions, it was highest in the Westnile and Karamoja regions - some of the most rural and poorest communities in the country. These two regions, however, still lag behind others in coverage of facility births. While this might be unexpected, it's not entirely surprising as these regions have received targeted support from many partners to improve maternal and newborn care. Several donors and development partners, while working with the Ministry of Health, have supported the maternal and newborn care services in these regions through strengthening both access and quality of care [58]. Previous studies in these areas have documented that while these donor-driven concerted efforts raise sustainability questions, they have been effective in improving health facility readiness [59,60]. This is a testament that targeted investment and commitment to strengthening facilities are achievable and necessary to bridge the quality gap [51,61]. It was also surprising that the Kampala region, Uganda's capital, had the lowest facility readiness. This aligns with previous studies, especially in Asia and Africa, which have indicated that while it is a commonly held assumption that urban facilities usually have better readiness than rural ones, this may not always be the case—particularly in areas where the population largely depends on free care from public or government facilities [62,63]. One possible explanation is that these facilities are often characterized by higher service volumes compared to those outside Kampala. The large number of births overwhelms the facilities, leading to stockouts of essential drugs and supplies, reduced functionality of equipment, strained referral systems, and shortages in human resources. Studies in Uganda have shown that urban and higher-level facilities are typically more affected by frequent stockouts and a lack of basic equipment due to high patient volumes [64,65]. There have been recommendations for collaborative action with district authorities within and around Kampala to equip surrounding facilities and streamline the referral system to address these challenges [66].

## Strengths and limitations

A key strength of this study is that we adjusted contact coverage for both readiness and content to give a more comprehensive view of effective coverage using two approaches. It is also one of the first studies in Uganda to use an effective coverage approach to assess the quality of maternal and newborn care. Further, while determining readiness scores, we used a more robust weighted additive method that minimizes under- or overestimation of readiness scores because of the varying number of readiness parameters in each indicator.

The study, however, had some limitations; there was a possibility of recall bias with the UDHS questionnaire collecting data on births in the three years preceding the survey. Although this doesn't eliminate the bias altogether, we restricted our analysis to the recommended recall period of two years and the most recent birth, as women are more likely to recall events of the most recent child. Another limitation of our study is that we used the most recent 2022 DHS survey data and the 2023 HHFA assessment data. This means that facility readiness was assessed after births in the DHS took place at least three years before the HHFA and so may not reflect the true readiness at the time of birth. There has, however, not been a SPA in Uganda since 2007, and the 2018 SARA excluded many facilities and had a high non-response, with only 135 facilities sampled nationally [67]. Although the HHFA data provides the exact names of facilities, the data doesn't provide exact places of birth for mothers for confidentiality and data protection reasons. As such, we could not link individual mothers to the exact place of delivery, which is the gold standard. We, however, used an ecological linking method by regions, and it has been shown that there is no significant difference in findings between the two methods [31,32]. Our study was limited by the scope of available indicators in UDHS data to determine the interventions received by mothers during childbirth. We therefore selected a set of interventions that may not have comprehensively represented all the recommended interventions. Additionally, the use of self-reported data could have potentially introduced recall and reporting bias.

## Conclusions

More than half of the women in our study did not give birth from ready health facilities, and more than four-fifths did not receive the standard content of care, indicating a very low effective coverage in Uganda. The lower government health centres had the lowest readiness. We recommend that future Maternal and Newborn programs and initiatives to improve maternal and newborn care from both government in Uganda and the donor partners should look beyond the demand-side challenges of utilisation of maternal health services. These programs should also strengthen the supply bottlenecks, such as human resources for health, drugs, supplies, and equipment. The effective coverage gaps were more apparent in lower government health centres that deliver the bulk of the facility-based birth care. We therefore also recommend targeted strengthening of the readiness in these lower-level facilities and upgrading facilities providing childbirth services to CEmONC standards. Improving the quality of care in these lower-level government facilities also helps reduce the costs patients incur seeking these services in private facilities. There is also a need for more effective supervision approaches to ensure the delivery of good quality care, especially in private facilities.

## Supporting information

**S1 Table. Harmonisation of regions and facility levels for linking DHS and HHFA data sets.**
(DOCX)

**S2 Table. Indicators in each domain used to measure the readiness-adjusted coverage of maternal and newborn care from HHFA data.**
(DOCX)

**S3 Table. Summary of DHS data Variables used and their analytical categories.**
(DOCX)

## Acknowledgments

We acknowledge the Institute of Tropical Medicine (ITM) for allowing BT to attend the course on "writing a paper based on the Demographic and Health Survey (DHS) data on reproductive and child health". We acknowledge the support of Anteneh Asefa in conceptualising this research. We furthermore extend our appreciation to the UBOS and Uganda's Ministry of Health for providing us with the data sets for this study.

## Author contributions

**Conceptualization:** Brian Turigye, Edgar Mugema Mulogo, Joseph Ngonzi, Peter M Macharia, Aliki Christou, Lenka Beňová.

**Data curation:** Brian Turigye, Miriam Acheng, Aliki Christou, Lenka Beňová.

**Formal analysis:** Brian Turigye, Peter M Macharia, Aliki Christou, Lenka Beňová.

**Funding acquisition:** Brian Turigye, Joseph Ngonzi.

**Investigation:** Brian Turigye.

**Methodology:** Brian Turigye, Edgar Mugema Mulogo, Joseph Ngonzi, Peter M Macharia, Aliki Christou, Lenka Beňová.

**Project administration:** Brian Turigye.

**Resources:** Brian Turigye, Joseph Ngonzi, Miriam Acheng, Lenka Beňová.

**Software:** Brian Turigye, Miriam Acheng.

**Supervision:** Edgar Mugema Mulogo, Joseph Ngonzi, Peter M Macharia, Aliki Christou, Lenka Beňová.

**Validation:** Miriam Acheng, Aliki Christou, Lenka Beňová.

**Visualization:** Brian Turigye, Aliki Christou, Lenka Beňová.

**Writing – original draft:** Brian Turigye.

**Writing – review & editing:** Brian Turigye, Edgar Mugema Mulogo, Joseph Ngonzi, Peter M Macharia, Miriam Acheng, Aliki Christou, Lenka Beňová.

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
