## [Decision Letter · Decision Letter 0]

19 Aug 2025

PGPH-D-25-01831

Beyond Facility-based Births: Is Uganda Delivering Effective Maternal and Newborn Care? An analysis of the 2022 Demographic Health Survey and 2023 Harmonized Health Facility Assessment Survey

Dear Dr. Turigye,

Thank you for submitting your manuscript to PLOS Global Public Health. After careful consideration, we feel that it has merit but does not fully meet PLOS Global Public Health’s publication criteria as it currently stands. Therefore, we invite you to submit a revised version of the manuscript that addresses the points raised during the review process.

We look forward to receiving your revised manuscript.

Kind regards,

Sham Lal

Academic Editor

Journal Requirements:

i. State the initials, alongside each funding source, of each author to receive each grant.

ii. State what role the funders took in the study. If the funders had no role in your study, please state: “The funders had no role in study design, data collection and analysis, decision to publish, or preparation of the manuscript.”

2. Please ensure that your Ethics Statement is available in its entirety at the beginning of your Methods section, under a subheading 'Ethics Statement'.

3. Please provide separate figure files in .tif or .eps format.

4. We have noticed that you have uploaded Supporting Information files, but you have not included a list of legends. Please add a full list of legends for your Supporting Information files after the references list.

5. We note that you have indicated that there are restrictions to data sharing for this study. For studies involving human research participant data or other sensitive data, we encourage authors to share de-identified or anonymized data. However, when data cannot be publicly shared for ethical reasons, we allow authors to make their data sets available upon request. For information on unacceptable data access restrictions, please see http://journals.plos.org/plosone/s/data-availability#loc-unacceptable-data-access-restrictions. 

Reviewers' comments:

Reviewer's Responses to Questions

**Comments to the Author**

1. Does this manuscript meet PLOS Global Public Health’s publication criteria?

Reviewer #1: No

Reviewer #2: Partly

2. Has the statistical analysis been performed appropriately and rigorously?

Reviewer #1: No

Reviewer #2: I don't know

3. Have the authors made all data underlying the findings in their manuscript fully available (please refer to the Data Availability Statement at the start of the manuscript PDF file)?

Reviewer #1: No

Reviewer #2: Yes

4. Is the manuscript presented in an intelligible fashion and written in standard English?

Reviewer #1: Yes

Reviewer #2: Yes

Reviewer #1: Dear authors,

I appreciate the opportunity to review this work; it is excellent to see continued work to make use of the HHFA and DHS data to better understand health services and effective coverage. The article is timely and relevant. While the analysis has good potential, there are a number of inconsistencies and potential misinterpretations that make it difficult to understand exactly what was done and if the results reflect the methods described. There are also ways the analysis and findings can be better integrated with the existing effective coverage literature, including refining the specific research question addressed and making terminology more precise and consistent. I raise below major and other concerns for your consideration.

Major

1) Use of effective coverage cascade terminology and interpretation of findings

1a) In line 135, the order of coverage types provided does not follow the cascade steps, and it's not clear why they would be presented out of order. If the idea is to provide a few exemplars, then I would suggest putting them in order and providing a brief parenthetical definition; alternatively, replace with a more general description of the cascade from population in need through health service use to health outcomes experienced. More broadly, the introduction should justify the specific focus on input-adjusted and intervention-adjusted coverage in this manuscript and the reason that the calculation of each measure can add value to understanding MNH services in Uganda.

1b) While it does not become evident until the results section, the measures reported are not strictly within a cascade since intervention coverage is calculated independently of input-adjusted coverage. (Also note that both measures are referred to as effective coverage, while the guidance documents cited suggest using the more specific terminology whenever possible.) The use of intervention coverage on its own without inclusion of readiness should be justified in the methods, with reference to how the use of these two types of measures provides particular insights - why calculate input-adjusted coverage at all if intervention coverage is conceptualized separate from readiness? Better integration between the two sets of findings would be helpful even if the nested coverage outcome is not calculated - how does knowing which regions have higher readiness but lower intervention coverage, or vice versa, help to target interventions?

1c) The difference in readiness between facilities of all types in Kampala vs other regions is quite striking, and surprising given that urban / capital areas often have higher infrastructure and human resources than outlying areas. Is this finding consistent with other studies of the Ugandan health system? Does it have face validity for the author team? Were there indicators that were systematically different in Kampala vs other regions? The findings around deficits in Kampala and the discrepancy between low readiness and higher relative intervention coverage should be interpreted and discussed on together.

1d) Line 426, the statement ‘The findings also indicate that along the cascade of effective coverage, the largest gap is in the content of care being provided’ is a bit misleading in that 1) the estimates presented are each independent of each other, so it’s not directly a cascade, and 2) the measure used for intervention coverage was more stringent than readiness coverage (all interventions received vs. average readiness score). Presumably if the indicator for facility readiness required all readiness items in place, those scores would drop substantially. The choice to operationalize the measures differently can be justified, but I would keep it in mind - and consider the validity of self-report for intervention indicators - when interpreting the findings.

2) Research aim / gap in knowledge: Line 148: The statement on prior research (lack of) regarding MNH effective coverage in Uganda is fairly broad and perhaps not entirely accurate without further qualification. There are certainly closely related studies related to delivery and newborn care (https://pubmed.ncbi.nlm.nih.gov/34967928/, https://pubmed.ncbi.nlm.nih.gov/39151985/) as well as multiple studies addressing ANC effective coverage, some of which describe this as maternal and newborn care (https://pubmed.ncbi.nlm.nih.gov/28581379/, https://pubmed.ncbi.nlm.nih.gov/30322649/, https://pubmed.ncbi.nlm.nih.gov/29632704/) or that focus on methods (https://pubmed.ncbi.nlm.nih.gov/29497508/). Several of these are already cited; I would suggest providing a more specific statement on the gap in the literature - e.g., specific to delivery care, applicable at the national level - so that it is easier to verify this claim.

3) Methodological clarity: there are a number of gaps and inconsistencies, particularly between the methods section and table headers and footnotes, that make it difficult to tell which procedures were followed and if the findings are accurately presented based on the methods described.

3a) The terminology of parameters within indicators is not standard; the DHS analyses and most quality of care analyses would refer to indicators within domains (e.g., adequate water supply within infrastructure domain).

3b) The weighting approach as described in the Methods section does not seem to align with Mallick et al’s weighted average. The Mallick et al approach assigns equal weights by domain regardless of the number of items per domain, in essence averaging first within domain and then across domains. The approach described here weights each domain by the number of items in it, essentially returning the readiness index to a simple average of all items included. Then in Table 2, two different approaches to defining readiness are described - actually weighting domains evenly (footnote) and using a binary indicator of all readiness items met (column header). Which was actually used (or at least, which was intended to be used)? Please define the actual method used in full in the Methods section, report the corresponding results, and reference back to the selected method only as necessary in the Tables.

3c) Line 292 - analysis relied on weighted data. More detail is required here: did it also incorporate sampling strata? Was the subpop option used to reflect the fact that the original sample is broader than the population analyzed here? Was weighting used for the HFA as well as DHS data? How were weights utilized in the HFA data, both for regional summaries like Table 2 and for the ecological merging within strata? Typically weights used for summarizing nationally or regionally are not appropriate for use within strata unless they are appropriately re-scaled.

3d) The methods section does not define the approach for calculating intervention coverage, which it is clear from the results section was estimated independently from the readiness adjusted coverage, making it not exactly a nested cascade as was described previously. This may be a justifiable approach, but requires description and justification in the methods. In general, while it’s helpful to have footnotes on tables for clarity, this information should not add any details that are not reported in the Methods text, but simply express the core idea (e.g., weighted N, or Intervention coverage = crude coverage x self-reported intervention receipt) if needed to orient readers

3e) The methods section states that an ecological linkage was conducted based on strata of facilities, but the Results section reports the direct product of regional crude coverage and regional readiness as the input-adjusted coverage figure, without any apparent incorporation of facility type. What happened to the facility strata? Using the figures provided in previous tables, input-adjusted coverage in Kampala would be 40.1% and mid-east would be 50.2% - not huge differences from the numbers shown, but still different.

Other

1) Please update the data availability statement to provide contact information for the request of this information by other research teams. It would also be possible to make fully public the simplified datasets including the readiness indicators and scores and the individuals' facility types.

2) Line 113 - please provide dates for the MMR estimates

3) Line 220: Is there a conceptual or empirical justification for using only most recent of multiples? Would slightly bias effect estimates relative to a target population of births in past 2 years.

4) Line 297 - I’m not sure that multiplying readiness score by crude coverage needs to be considered a ‘WHO formula’ given that it is a standard component to effective coverage analyses; if there’s a specific need to reference the WHO formula, the accompanying citations should provide this directly, rather than representing 2 of the many EC analyses that have taken this approach.

5) Citation for the delta approach? Please note there is publicly available code for calculating stratified regional effective coverage estimates with variance options that could be applied for this analysis: https://pmc.ncbi.nlm.nih.gov/articles/PMC7101480/

6) Line 319 - 321: 20 - 34 years is a broad swatch of child-bearing women, not sure there’s much value in such a coarse classification unless the goal is to implicitly identify the proportion of women in potentially higher risk adolescent and 35+ age groups, though this could be better achieved by directly stating these %s.

7) Table 1: please include in the table description or footnotes that figures represent weighted data and (presumably) that counts have been rounded to nearest whole number

Small query, but I was surprised that 97% of births were not intended - I don’t have the data to check, but the 2016 DHS findings indicated that 59% of births in the past 5 years were wanted at the time of conception. Just curious at this finding and the definition of intended used here to confirm.

8) Table 2: To what does the cross-symbol footnote on ‘unweighted percentage availability’ refer? I do not see the symbol in the table or know how to interpret this note.

9) Line 457 - citation?

10) Line 422: 86% of women were not lost between crude and intervention - 86% of women did not receive intervention coverage. The drop from 86% crude coverage to 14% intervention coverage is a linear difference of 72% and a relative difference of 84%, though I'm not sure adding additional %s to the interpretation is particularly helpful!

11) Lines 450 - 466: note there is ongoing discussion of the extent to which government health centers can be reasonably equipped for delivery care and the wisdom of attempting this vs. centralizing delivery care (for instance:https://pubmed.ncbi.nlm.nih.gov/33055093/, https://pubmed.ncbi.nlm.nih.gov/33055095/). This doesn’t have to be a focus of the discussion section, but the findings that such major gaps persist in government health centers does contribute to the broader discussion.

12) Citation 34 and 44 should be formatted as a journal article, coming through as an internet source

Reviewer #2: Overarching comments

The issues being addressed – effective coverage for maternal newborn care is an extremely important one.

Looking at the title however, and emphasis on ‘facility-based births’, mentioned again in line 152, and ‘childbirth’ and ‘childbirth care’ - mentioned throughout the article, the 10 interventions that were selected (lines 276-280), do not sufficiently cover ‘childbirth care’ especially intrapartum care, a period when mothers and the fetus and newborn are at risk. This has a lot to do with the data sources that were available, used – DHS and HHFA. With that said, the limitation of these data visi a vis the objective, although stated, is not sufficiently acknowledged. In addition, covering other outcomes in particular stillbirths, within limits of data availability and quality, would have added a compelling narrative on quality of care which is the main argument the paper is making, that could have been explored but wasn’t - approximately one half occur during the intrapartum period. Without the emphasis on facility-births alone, the authors could have explored additional indicators such as the coverage targets tracked by Every Woman Every Newborn Everywhere formerly ENAP/EPMM such as ANC4+, skilled health personnel at birth, postnatal care, possibly EmOC from the HHFA.

With that said the rest of the review addresses the authors’ submission as is.

Abstract: In the background ‘coverage’ in line 38 should be qualified as the paper is making a point to qualify the different kinds of ‘coverage’. Methods: Lines 53 and 54 (and 60) ‘all 10 recommended interventions’ – I think this has to do less with recommendations, than with data available, and should be qualified as such (related to the overarching comment above). The Conclusion does not sufficiently capture what was studied and is proposed.

Introduction: line 105 – should this be ‘newborn’ rather than ‘child’; and qualify SDG target 3.2 as newborn and under-five mortality (line 107). Line 111 – the first month of life is the most vulnerable for ‘newborns’ – did you mean ‘under-fives’? And stillbirths mentioned line 112, (and 117), reinforcing the point make in the overarching comment. Line 113 on maternal mortality – 70 per 100,000 live births being the global target and no country over 140 per 100,000 live births.

Results and discussion:

Table 1: very struck by the data on ‘pregnancy intention’ ….

Table 4: in effect the premise of the whole paper, is that having a facility birth (which is what is considered as ‘crude coverage’ means there is a skilled health worker with the resources to carry out the 10 interventions but it is not well explained, and in fact for two of the interventions, the data may also be capturing women who delivered at home and their newborns (post-natal care coverage)

Line 387 ‘ten care interventions for childbirth’ – same comment as before

Lines 430-436 – in fact the intervention coverage as presented, is also counting ‘contact’ and not ‘content’. For example, several studies have remarked the low quality of counseling that mothers receive – that would have been useful to explore further and/or acknowledge the limitation

There are other findings and opportunities that are not sufficiently highlighted/picked up in the discussion such as the encouraging findings for Govt. hospitals, the gaps in readiness for private health facilities – a very significant provider and potential to improve care for millions who are accessing care at high cost at these facilities

**Do you want your identity to be public for this peer review?** For information about this choice, including consent withdrawal, please see our Privacy Policy

Reviewer #1: No

Reviewer #2: No

---

## [Editor Report · Decision Letter 1]

13 Oct 2025

Beyond Facility-based Births: Is Uganda Delivering Effective Maternal and Newborn Care? An analysis of the 2022 Demographic Health Survey and 2023 Harmonized Health Facility Assessment Survey

PGPH-D-25-01831R1

Dear Dr Turigye,

We are pleased to inform you that your manuscript 'Beyond Facility-based Births: Is Uganda Delivering Effective Maternal and Newborn Care? An analysis of the 2022 Demographic Health Survey and 2023 Harmonized Health Facility Assessment Survey' has been provisionally accepted for publication in PLOS Global Public Health.

Best regards,

Sham Lal

Academic Editor